# EPANet-KD: Efficient progressive attention network for fine-grained provincial village classification via knowledge distillation

**Cheng Zhang**[1,2], **Chunqing Liu**[1,2]*, **Huimin Gong**[1,2], **Jinlin Teng**[1,2]

**1** College of Landscape Architecture and Art, Jiangxi Agricultural University, Nanchang, China, **2** Jiangxi Rural Culture Development Research Center, Nanchang, China

* liuchunqing@jxau.edu.cn

## Abstract

### Objective

Fine-grained classification of historical traditional villages plays a crucial role in guiding the future development and construction of urban and rural areas. This study aims to propose a new dataset for fine-grained classification of traditional villages and to propose an efficient progressive attention network for the problem of low accuracy and efficiency of fine-grained traditional historical village classification.

### Methods and results

Firstly, in order to further study the long-standing problem of fine-grained classification of traditional villages, a new fine-grained classification dataset of traditional villages containing 4,400 images, referred to as PVCD, is proposed by crawling and hand-arranging. Secondly, a new Progressive Attention Module, abbreviated as PAM, is also proposed. PAM engages in attentional modeling of prominent spatial features within the spatial dimension, subsequently applying attentional modeling to channel features beneath the identified salient spatial features. This process involves salient spatial feature attention modeling of prominent channel features within the dimension to extract discriminative information for fine-grained classification, thereby enhancing the performance of classifying traditional villages with precision. Finally, a new knowledge distillation strategy of softened alignment distillation, or SAD for short, is proposed, which simply and efficiently transfers the knowledge of softened category probability distributions through. Notably, based on the above proposed PAM, the lightweight EPANet-Student and the heavyweight EPANet-Teacher are proposed. In addition, the heavyweight EPANet-Teacher transfers the knowledge of fine-grained categorization of traditional villages to the lightweight EPANet-Student through the proposed SAD, abbreviated as EPANet-KD. The experimental results show that the proposed EPANet-Teacher achieves state-of-the-art performance with an accuracy of 67.27%, and the proposed EPANet-KD achieves comparable performance to the proposed EPANet-Teacher with 3.32M parameters and 0.42G computation.

**Data Availability Statement:** https://github.com/Jack13026212687/EPANet-KD.

**Funding:** Author: Cheng Zhang, Huimin Gong, Jinlin Teng. Corresponding author: Chunqing Liu. National Natural Science Foundation of China, for

the project: Gene Identification and Map Construction of Traditional Rural Settlement Landscapes in the Ganjiang River Basin (Serial No. 5196080626); Chunqing Liu:Head of the National Natural Science Foundation and responsible for the validation of the effectiveness of the innovation point.Cheng Zhang: research design, data collection and analysis, manuscript writing. Huimin Gong: study design, data collection and analysis, decision to publish. Jinlin Teng: study design, data collection and analysis, decision to publish.

**Competing interests:** The authors have declared that no competing interests exist.

## Conclusion

The proposed EPANet-KD maintains a good balance of accuracy and efficiency in the fine-grained classification of traditional villages, considerably promoting the research on the fine-grained classification of traditional villages. In addition, it facilitates the digital preservation and development of traditional villages. All datasets, codes and benchmarking results are publicly available for the promotion of this research area. https://github.com/Jack13026212687/EPANet-KD.

## Introduction

Traditional villages cover natural ecological space, agricultural production space and human living space, and are considered spatial carriers of rural residents interacting with natural, social, economic, and cultural environments [1]. According to Liu et al. [2, 3], Chinese traditional villages refer to ancient towns, ancient towns, and ancient cities, with long cultural heritage, complete historical characteristics, and rich intangible cultural and material relics, which not only maintain the characteristics of the "context", but also unique historical and cultural memories as well as landscape characteristics, although the form of settlement has changed accordingly. The settlement form has changed accordingly. Ding et al. [4, 5] pointed out that traditional villages have the following four characteristics: the first is a profound historical and cultural accumulation, which has been completely continued; the second is a rich intangible cultural heritage; the third is a basically intact rural pattern; and the fourth is the unique local customs and habits with unique regional characteristics. Traditional villages are one of the unique ideological carriers of the Chinese nation, the crystallization of the wisdom of the people of all nationalities, and the expression and inheritance of the excellent traditional Chinese culture [6, 7]. In general, Chinese traditional villages are spatial carriers of rural residents interacting with natural, social, economic, and cultural environments, with rich historical and cultural accumulations, intangible cultural heritage, intact rural patterns and unique local customs, which are the unique ideological carriers of the Chinese nation and an important manifestation and inheritance of Chinese traditional culture. Mast et al. [8] employed a full convolutional neural network to generate an urban villages map. Yu et al. [9], leveraging multi-modal intelligent computing and deep neural networks, focused on constructing a landscape design system. Buscombe et al. [10] applied deep neural networks for landscape classification, while Mougiakakou et al. [11] utilized neural network technology for landscape perception classification, and Liu et al. [12] employed chained fully convolutional neural networks to achieve accurate building extraction from fused DSM and UAV images. However, these deep learning-based approaches failed to concentrate much on the traditional village fine-grained classification task. To this end, a new traditional village fine-grained classification dataset called PVCD is hereby proposed to promote research in this area.

In recent years, with the rapid development of deep learning, an increasing number of classification networks have been proposed. For instance, Szegedy et al. [13] proposed a deep convolutional neural network architecture that keeps the computational budget unchanged by increasing the depth and width. Simonyan et al. [14] evaluated networks with increased depth and found that depths up to 16–19 layers significantly improve classification performance. He et al. [15] proposed a residual learning framework that prevents the training gradient from disappearing in deep networks. Xie et al. [16] proposed a transformer to unify the transformer with a lightweight multilayer perceptron decoder. To address the problem of scale variation

and high-resolution differences, Liu et al. [17] proposed a layered transformer to flexibly model different scales. Dosovitskiy et al. [18] proposed a ViT architecture that combines attention with a convolutional network or replaces some of the components. Liu et al. [19] redesigned the ViT to test the limiting performance of purely convolutional neural. Yu et al. [20] emphasized that the overall architecture of the transformer is more critical to performance, rather than specific modules, and Nagrani et al. [21] proposed a transformer-based architecture using a "fusion bottleneck" for multilayer modal fusion. In general, the network depth and performance have been continuously improved through deep convolutional neural networks, residual learning, transformers and other architectures and model design, and considerable attempts have been made to solve the challenges of training difficulties, scale variations, and high-resolution differences, with ViT and transformer-based frameworks becoming an important direction of exploration. Guo et al. [22] introduced an external attention mechanism to address the quadratic complexity of self-attention through two small learnable shared memories. Qin et al. [23] addressed the problem of information loss in the channel attention mechanism through frequency analysis. Liu et al. [24] proposed a global attention mechanism emphasizing information retention across dimensional interactions. Guo et al. [25] proposed a target-aware Siamese graphical attention network to improve the global matching problem of target feature regions. Yang et al. [26] proposed a simple but effective attention module for convolutional neural networks. Brauwers et al. [27] proposed a comprehensive framework to categorize various attention models. Pan et al. [28] found a potential relationship between convolution and self-attention. Zhang et al. [29] proposed a multi-branching architecture that applies channelized attention to the improvement of convolutional neural network representation learning. Guo et al. [30] introduced large kernel attention to accommodate the two-dimensional nature of images and improve the long-term relevance of self-attention.Hu et al. [31] adaptively recalibrate the feature responses in the direction of the channels by explicitly modeling the interdependencies between the channels. Guo et al. [32] introduce a spatial attention module, which infers the attention map along the spatial dimensions and multiplies the attention map with the input feature maps for adaptive feature refinement. Zhu et al. [33] explore the cross-layer features with similar semantic features between the intrinsic connections, proposed a feature-semantic fusion module to enhance the diversity of global feature information. Kuang et al. [34] proposed a multi-scale global-local semantic graph network for multi-label image classification considering the overall features of the image and accurately locating the target region. Yang et al. [35] proposed a graph-relational decision network in order to establish the relationship between the targets. Collectively, these researchers have contributed significantly to the evolution of convolutional neural network representation learning, and have achieved innovations including the integration of external attention mechanisms, frequency analysis, global attention mechanisms, target-aware networks, and streamlined yet efficient convolutional neural network attention modules. These advancements address challenges such as the complexity of self-attention, information loss, cross-dimensional interactions, and global matching. However, existing state-of-the-art classification networks, while demonstrating advanced performance, tend to overlook the comprehensive exploration of spatial and channel dimensional attention in high-level semantic information. This integrated exploration is particularly crucial for addressing the requirements of fine-grained classification tasks. To this end, a progressive attention module called PAM is hereby proposed. Unlike advanced attention modules, the proposed PAM directs attention to salient features in space in the spatial dimension, followed by attention modeling to salient channel features under the channel dimension based on the salient spatial features. This approach aims to extract discriminative information for fine-grained classification, ultimately enhancing the performance of traditional village fine-grained classification.

Knowledge distillation is a simple but effective deep model compression technique that aims to transfer knowledge learned from large teacher models to small student models. In recent years, numerous knowledge distillation strategies have been proposed. For instance, Chen et al. [36] proposed a framework to learn compact and fast target detection networks through knowledge distillation and cue learning to improve accuracy. In order to extract knowledge from weakly-supervised detection tasks to improve multi-label classification, Liu et al. [37] proposed deep framework and investigated knowledge distillation strategies, including pixel-level distillation, to apply distillation schemes for image classification to semantic segmentation. Xu et al. [38] explored knowledge distillation methods to extract richer knowledge from teacher models. In order to mitigate overfitting, Yun et al. [39] proposed a regularization method to penalize the predictive distribution between similar samples. Ho et al. [40] addressed anomaly classification of chest X-ray images using knowledge distillation to solve the challenge of high quality interpretation and reporting of medical images. Zhang et al. [41] explained the success of knowledge distillation from an information theoretic perspective to quantify the knowledge of the DNN middle layer. In order to distill the performance of the integration into a single model, Allen-Zhu et al. [42] analyzed the case of a homogeneous neural network integrating output averaging to obtain the best performance by training with the same algorithm. Collectively, these researchers have provided solutions addressing overfitting concerns and extracting richer knowledge within a knowledge distillation framework. Their work spans diverse domains such as target detection, multi-label classification, semantic segmentation, and medical image interpretation, and the overarching goal is to cultivate compact, fast models with enhanced performance through the strategic application of regularization and integration methods. However, existing state-of-the-art classification networks are inefficient in their inference, and thus unusable for resource-constrained end devices. Inspired by the fact that knowledge distillation can compress networks to fully balance accuracy and efficiency, the SAD knowledge distillation strategy is proposed. Unlike state-of-the-art knowledge distillation strategies, the simple and efficient transfer of traditional village fine-grained classification knowledge from EPANet-Teacher to EPANet-Student is facilitated by softening the alignment of fine-grained classification output probability distributions.

The main contributions are as follows:

1. First, the first new traditional village classification dataset called PVCD is constructed, which contains 4,400 images of different villages in Jiangxi Province, and divides Jiangxi Province into 11 different regions for classification and labeling. In addition, the constructed PVCD dataset has been open-sourced, thus facilitating the flourishing of deep learning-based models in the traditional village research community.

2. Second, a new progressive attention module, or PAM for short, is proposed, which directs attention to salient features in space in the spatial dimension, followed by attention modeling based on salient spatial features under salient channel features in the channel dimension. In this case, discriminative fine-grained classification information is obtained, and the performance of fine-grained classification of traditional villages is improved.

3. Third, a new soft-aligned knowledge distillation strategy, or SAD for short, is hereby proposed. Specifically, this strategy facilitates the simple and efficient transfer of traditional village fine-grained categorization knowledge from EPANet-Teacher to EPANet-Student by soft-aligning the category probability distributions of fine-grained categorization outputs.

4. Finally, it is demonstrated through extensive experimental results that the EPANet-Teacher and EPANet-KD proposed in this paper exhibit the best performance among the existing

state-of-the-art methods. Notably, EPANet-KD achieves a sufficient balance of accuracy and efficiency with 3.32M parameters and 0.42G computation.

## Materials and methods

### PVCD dataset

Herein, a new PVCD is proposed to facilitate the research on the task of fine-grained categorization of traditional villages. Fig 1 illustrates the topography, geomorphology, and kernel density analysis of the distribution of villages in Jiangxi Province, and Fig 2 presents an example of an image used for fine-grained categorization of traditional villages in Jiangxi Province, where each column represents an area category. There are 11 region categories in total, and the names of these region categories are Fuzhou, Ganzhou, Ji'an, Jingdezhen, Jiujiang, Nanchang,

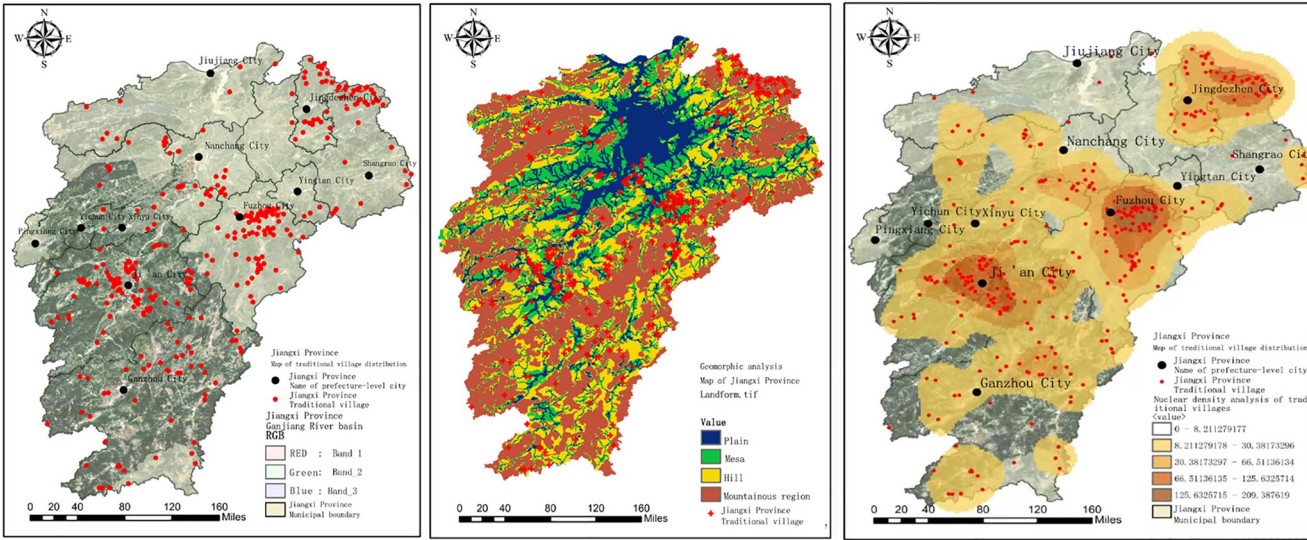

**Fig 1. Topography, geomorphology and kernel density analysis of village distribution in Jiangxi Province.**

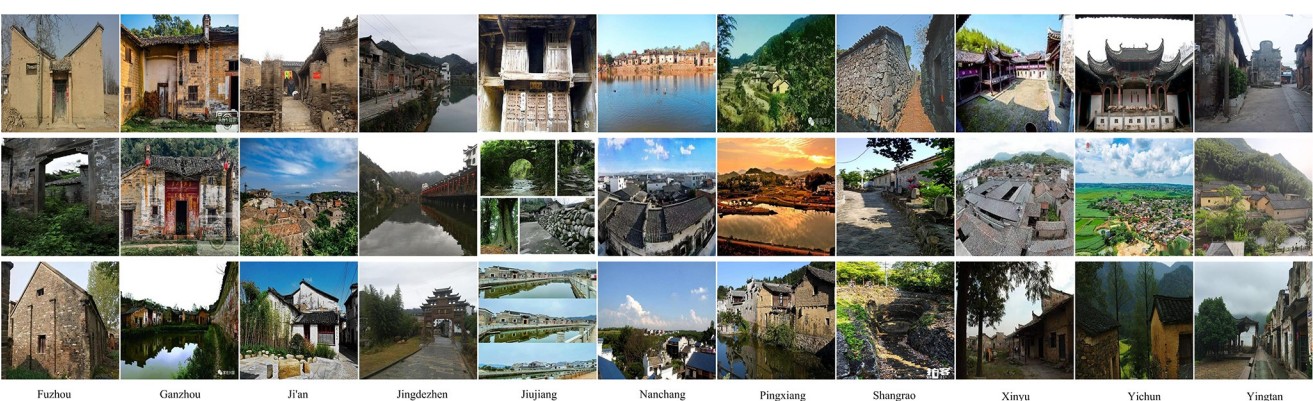

**Fig 2. Example of the proposed fine-grained categorized images of traditional villages in Jiangxi Province, with each column representing a regional category.**

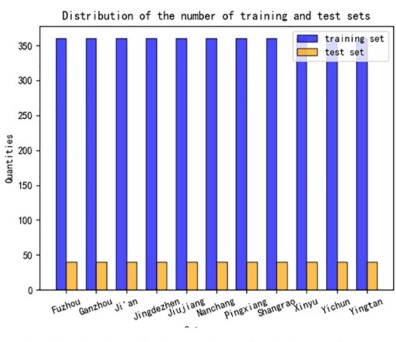
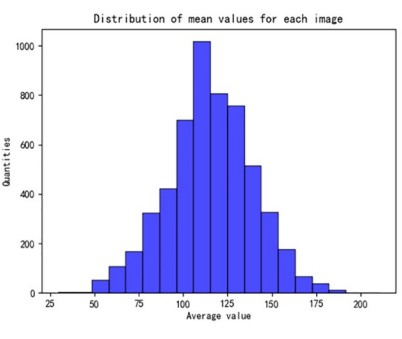
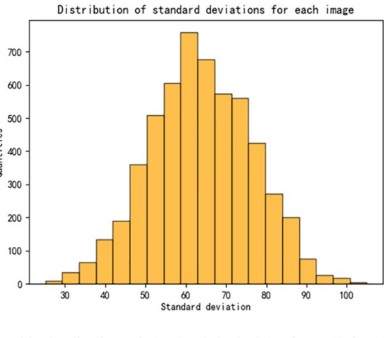

(a)Distribution of the number of training and test sets  (b)Distribution of mean values for each image  (c)Distribution of standard deviations for each image

**Fig 3. Histogram of the distribution of image data volume, image gray level mean and image standard deviation distribution of the proposed fine-grained classification of traditional villages in Jiangxi Province.**

Pingxiang, Shangrao, Xinyu, Yichun, and Yingtan, respectively. As shown in the image examples, each region possesses its own characteristic style, and since they all belong to Jiangxi Province, a great similarity can be observed between each region. Fig 3 shows the histogram of the distribution of data volume, the histogram of the distribution of image gray scale mean and image standard deviation for the proposed fine-grained classification image of traditional villages in Jiangxi Province. It can be seen from Fig 3(a) that in order to prevent the deep learning algorithm from the long-tail effect, the number of training and test sets for each regional category is controlled at a ratio of 9:1, where there are 360 samples for each category in the training set and 40 for each category in the test set. As shown in Fig 3(b), the gray scale mean value of each image is concentrated between 75–150, indicating a great similarity between these images. In addition, it can be seen from Fig 3(c) that the grayscale standard deviation of each image is concentrated between 50–85, revealing very similar differences between these images, further indicating the essential role of a new proposed PAM in improving the fine-grained classification of traditional villages.

## Overview

In order to improve the performance of efficient fine-grained classification of traditional villages, EPANet-KD is hereby proposed, the general structure of which is shown in Fig 4. It consists of four components: the Segformer [38] encoder, PAM, SAD distillation strategy, and total loss.

**A. Segformer encoder.** Different from existing purely CNN-based classification methods that extract features in a local sliding window, the EPANet model uses Segformer [43] as an encoder backbone to exploit the global cues embedded in the input patches in a convolution-free manner, a backbone that overcomes the limitations of simple sliding windows in CNNs. Specifically, Inspired by [38], a very deep heavyweight backbone network is chosen for high accuracy performance, and a shallow lightweight backbone network is chosen for efficient inference performance. Thus,EPANet-Teacher adopts Segformer-b4 as the backbone, while EPANet-Student uses Segformer-b0 as the backbone, and both of them obtain four-layer outputs, respectively, with the only difference being the number of channels in Segformer-b4. The channels encoded in Segformer-b4's four-layer encoding are 64, those of Segformer-b4 are 64, 128, 256 and 512, denoted by $\{X_{Ti}|i = 1, 2, 3, 4\}$, and those of Segformer-b0 are 32, 64, 128 and 256, denoted by $\{X_{Si}|i = 1, 2, 3, 4\}$.

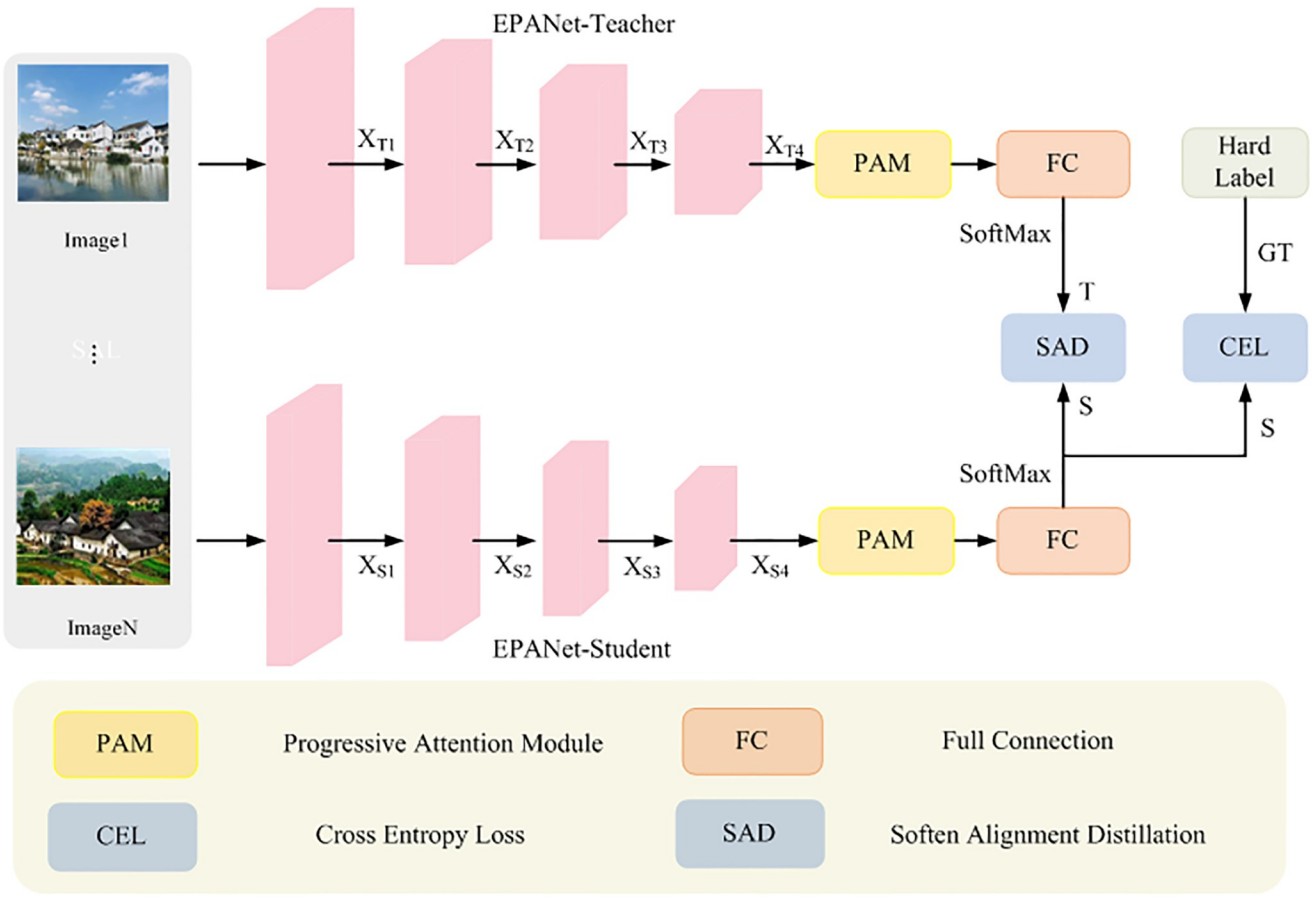

**Fig 4. General structure of the proposed EPANet-KD.**

**B. PAM.** Existing state-of-the-art networks fall short in exploring attention across both spatial and channel dimensions. There is a pressing need for a more discriminative module to solve this limitation, particularly for fine-grained categorization of traditional village images, which often exhibit significant similarity. Taking inspiration from the attention mechanism in the human brain, this paper introduces a Progressive Attention Module. The module is designed to enhance the performance of traditional village fine-grained classification by intricately modeling both spatial and channel dimensions, thereby facilitating the extraction of discriminative information crucial for accurate categorization. As shown in Fig 5, specifically, the most salient information on each channel layer is first obtained by maximum pooling along the spatial dimension, and then, these salient information are nonlinearly transformed by a 1x1 convolution kernel. Afterwards, the less salient information is filtered by a Relu activation function. Secondly, in order to reduce the difference caused by the direct mapping of salient information, the salient information is compressed into less salient information in two steps, i.e., from the salient information in 512 dimensions to 256 dimensions, and then from the salient information in 256 dimensions to 128 dimensions. Then, the most salient information in 128 dimensions is mapped to the salient information in 512 dimensions in reverse two-steps to perform the sigmoid activation operation to obtain the spatial saliency attention map $A_S$. $A_S$ is obtained, and broadcast multiplication is performed with the original input and then summed for the salient feature information along the spatial dimensions. Finally, the salient

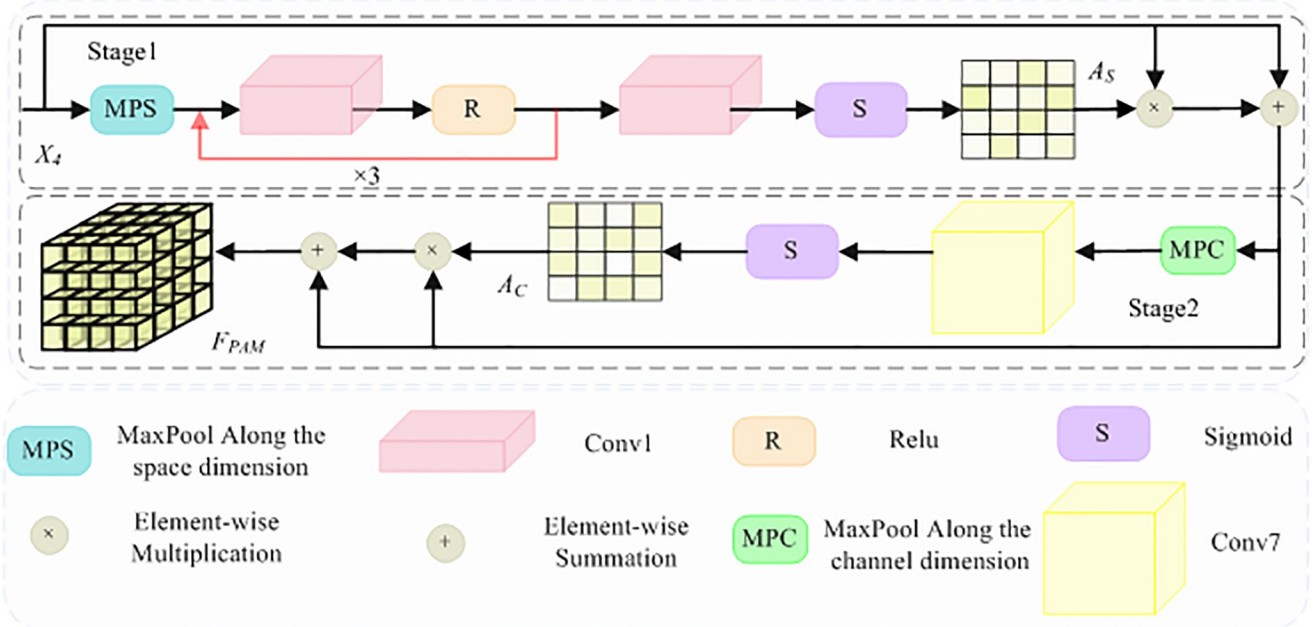

**Fig 5. Proposed Progressive Attention Module (PAM).**

spatial features are maximally pooled along the channel dimension, and a Sigmoid activation operation is performed to obtain the channel saliency attention map $A_C$, which is multiplied and summed with the salient spatial features again to obtain the salient spatial and channel attention features $F_{PAM}$. Their formulas are specified as follows:

$$A_S = S(Conv1(R(Conv1(R(Conv1(R(Conv1(MPS(X_4))))))))) \tag{1}$$

$$A_C = S(Conv7(MPC(X_4 \times A_S + X_4))) \tag{2}$$

$$F_{PAM} = (X_4 \times A_S + X_4) \times (A_C + 1) \tag{3}$$

where, $Conv_1$ denotes a $1 \times 1$ convolutional kernel with a step size of 1; $R$, $Relu$ activation function operation; $S$, the $Sigmoid$ activation function processing; $MPS$, maximum pooling along the spatial dimension; $MPC$, maximum pooling along the channel dimension; and $Conv_7$, the $7 \times 7$ convolution kernel with step size 7.

**C. SAD distillation strategy.** Existing advanced networks involve billions or tens of billions of parameters and thus require more computational resources for inference, making it difficult for these networks to use in resource-limited environments. In addition, the large number of parameters in extant advanced networks results in relatively slow inference, which limits their use in certain real-time applications or low-latency scenarios. To address these issues, inspired by knowledge distillation, a SAD knowledge distillation strategy is hereby proposed. Traditional knowledge distillation involves direct supervision of knowledge transfer through category-by-category probability, which greatly increases the difficulty for teachers to impart knowledge and for students to accept it. Different from other traditional knowledge distillation, this strategy facilitates the simple and efficient transfer of EPANet-Teacher's traditional village fine-grained categorization knowledge to EPANet-Student by softening the

alignment of the category probability distributions of the fine-grained categorization outputs, a process in which the EPANet-Student being taught is called EPANet-KD. The SAD's knowledge distillation strategy formula is specified as follows:

$$S = SoftMax(S/11.0) \tag{4}$$

$$T = SoftMax(T/11.0) \tag{5}$$

$$SAD = \left( \sum\nolimits_{i=1}^{C} (S_i - T_i) \right)^{1/2} \tag{6}$$

where, $S$ denotes the classification probability output of EPANet-Student; $T$, the classification probability output of EPANet-Teacher; $SoftMax$, the normalization operation; $SAD$, the soft alignment distillation loss; and $C$, the number of fine-grained categories.

**D. Total loss.** Highly accurate EPANet-Teacher is obtained by training with $L_T$ as supervisor. In addition, in order to train an efficient and better accurate EPANet-Student, $Loss$ is used for the supervision. Their formulas are specified as follows:

$$L_T = -\sum\nolimits_{i=1}^{C} T_i log(GT_i) \tag{7}$$

$$CEL = -\sum\nolimits_{i=1}^{C} S_i log(GT_i) \tag{8}$$

$$Loss = SAD + CEL \tag{9}$$

where, $L_T$ denotes the loss of EPANet-Teacher; $CEL$, the cross-entropy loss; $Loss$, the total loss of EPANet-Student; and $GT$, the image true category label.

## Results

### A. Experimental environment

The experiments are conducted on a computer configured with an Intel i5-7500 processor and an NVIDIA Jumbo XP graphics card. The PyTorch library is used, and the parameters of the Segformer backbone network are obtained from models pre-trained on the ImageNet dataset. To optimize the network parameters, the Adam optimization algorithm is adopted. During training, the batch size was set to 30, and the initial learning rate was set to 1e-5. 300 training epochs are performed. For training and testing, the input RGB images are resized to $224 \times 224$ pixels. In order to enhance the diversity of the data, data enhancement techniques such as random flip, rotation and boundary cropping for all training images are used for data augmentation.

### B. Quantitative analysis

In order to further demonstrate the effectiveness of the method for fine-grained classification of traditional villages, the proposed network is compared with the state-of-the-art VGG19, GoogLeNet, ResNet152, ConvNeXt-Large, Swin-Big, Vit-Big, and MetaFormer classification networks in terms of the number of parameters, as shown in Table 1 and Fig 6 below, respectively. The number, computation, accuracy, ROC value and confusion matrix results are quantitatively analyzed for detailed experiments.

As shown in Table 1, it can be concluded that the ConvNeXt-Large, and Swin-Big methods fail to make accurate predictions, while the VGG19, GoogLeNet, ResNet152, Vit-Big, and

**Table 1. Quantitative comparison results, ↑ and ↓ indicate larger and smaller, respectively.**

| Method | Published Year | Params(M)↓ | Flops(G)↓ | Accuracy↑ | ROC↑ |
|---|---|---|---|---|---|
| VGG19 | 2014 | 139.62 | 19.63 | 63.86 | 0.89 |
| GoogLeNet | 2015 | 5.61 | 1.51 | 63.41 | 0.88 |
| ResNet152 | 2016 | 58.14 | 11.60 | 63.41 | 0.86 |
| ConvNeXt-Large | 2020 | 196.21 | 34.39 | 58.64 | 0.84 |
| Swin-Big | 2021 | 58.72 | 10.22 | 55.91 | 0.84 |
| Vit-Big | 2021 | 59.08 | 2.95 | 63.41 | 0.85 |
| MetaFormer | 2022 | 11.62 | 1.89 | 60.91 | 0.85 |
| EPANet-Student | - | 3.32 | 0.42 | 50.45 | 0.80 |
| EPANet-KD | - | 3.32 | 0.42 | 64.31 | 0.85 |
| EPANet-Teacher | - | 81.45 | 11.38 | 67.27 | 0.88 |

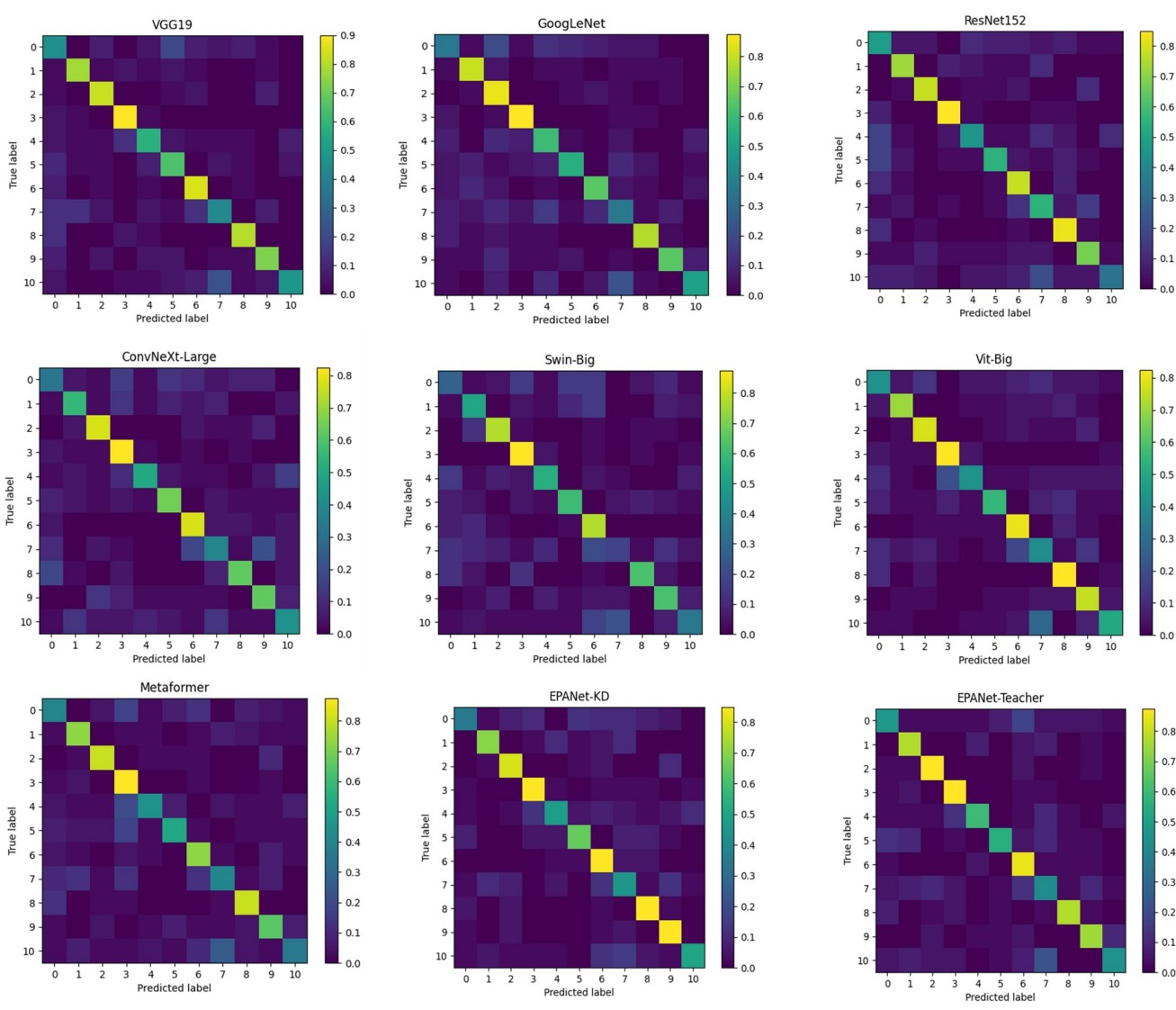

**Fig 6. Confusion matrix visualization results of the state-of-the-art method and the proposed method.**

MetaFormer methods have a larger number of parameters and Flops relative to our methods, although they are relatively accurate in predicting the results of the test set. This leads to a significant increase in computational and storage costs, making it unusable for resource-constrained end devices. In contrast, the proposed EPANet-Teacher method has the best accuracy and the second highest ROC value, thanks to the proposed PAM. More importantly, the proposed EPANet-KD method has the best number of parameters and Flops, and its accuracy and ROC value are close to those of the proposed EPANet-Teacher method, further proving the accuracy and ROC value of the proposed SAD knowledge distillation strategy.

It can be concluded from Fig 6 that more yellow color on the diagonal line represents higher prediction accuracy, and 0–10 in the figure represents Ji'an, Jiujiang, Yichun, Jingdezhen, Shangrao, Ganzhou, Nanchang, Yingtan, Fuzhou, Xinyu, and Pingxiang regions, respectively. First, it can be observed from the figure that the proposed EPANet-Teacher method has the best model generalization ability in the whole. Besides, the above network has better prediction results for all six categories including 1, 2, 3, 6, 8 and 9. Finally, by observing the diagonal colors of the EPANet-KD confusion matrix, it can be found that the proposed SAD simply and efficiently transfers teachers' fine-grained categorical knowledge to students.

## C. Qualitative analysis

In order to analyze the proposed EPANet-KD and other state-of-the-art classification networks from a qualitative point of view, the following Figs 7 and 8 represent the Grad-CAM [44] attention visualization and t-SNE [45] comparison graphs of the fine-grained classification network of traditional villages, respectively. Attention graphs and t-SNE can well measure the performance of the classification network from a qualitative point of view.

As shown in Fig 7, each row represents an example of an RGB traditional village image, the first column represents the original image, and each column, starting from the second column, represents a classification method. The yellow and red colors in the image indicate the feature information of the most characteristic region in the image. As a whole, the proposed method more fully focuses on the regions featured in each image, while other methods either do not focus enough on the featured regions or focus on other regions that are not featured, e.g., the sky. This further demonstrates the superiority of the proposed method.

As shown in Fig 8, the t-SNE tool can be used to map the high-dimensional invisible data representations to a two-dimensional intuitionistic space, where the same color indicates the same class. As can be seen from the figure, the proposed EPANet-KD is more compact within classes and has larger gaps between classes compared to GoogLeNet, ResNet152, and VGG19 methods. In addition, it can be seen from the last row in the figure that from the first column to the last column, the compactness between the same classes is getting higher, and the difference between different classes is getting bigger, which is attributed to the fact that the proposed EPANet-Student makes full use of the SAD knowledge distillation strategy to learn the effective fine-grained categorization knowledge of EPANet-Teacher.

## Discussion

In order to validate the effectiveness of the PAM module and the SAD knowledge distillation strategy proposed in this paper, ablation experiments are performed on each of them. In addition, we visualize and analyze the shortcomings of the proposed methods.

### A. Efficiency of PAM

First, traditional deep learning classification methods directly maximize the pooling of the deepest semantic information and then perform fully connected classification, resulting in

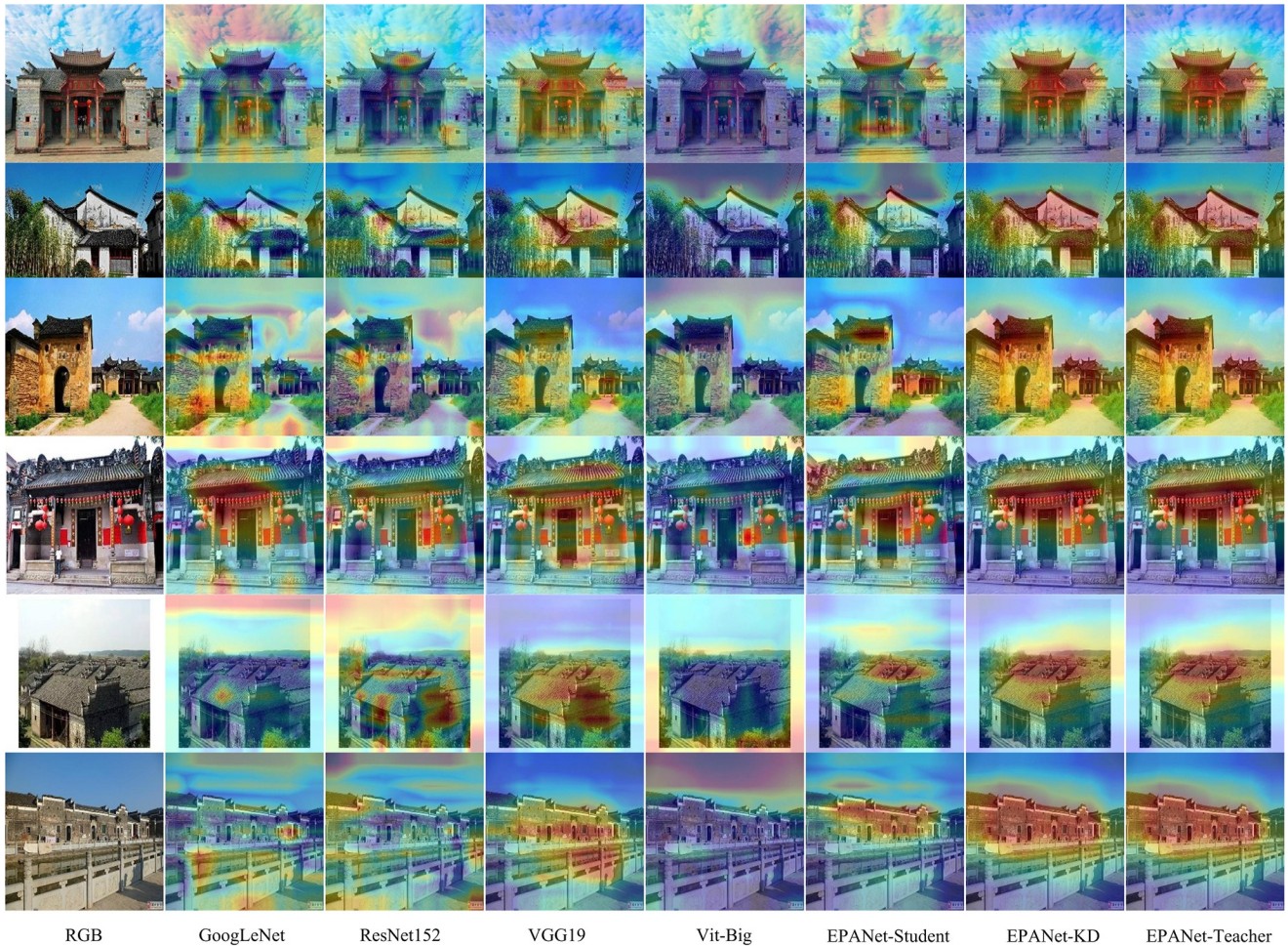

RGB          GoogLeNet          ResNet152          VGG19          Vit-Big          EPANet-Student          EPANet-KD          EPANet-Teacher

**Fig 7. Comparison of attention visualization of fine-grained classification networks for traditional villages.**

excessive invalid information affecting the final classification results. Second, the comprehensive exploration of channel and spatial information in high-level semantic information is insufficient. To this end, a PAM module that removes invalid information by gradually paying attention to the channel dimension and spatial dimension of high-level semantic information to obtain a network with stronger generalization ability is proposed, and the quantitative results and visualization results of the ablation experiments of the PAM are shown in Table 2 and subfigure a of Fig 9, respectively.

As depicted in Table 2, the notations W/O PAM, W/O Stage 1, and W/O Stage 2 refer to the absence of the PAM, the removal of the first stage, and the removal of the second stage in the PAM based on EPANet-KD, respectively. Table 2 reveals that EPANet-KD exhibits an improvement of 4.99 in Accuracy and 0.1 in ROC relative to EPANet-KD with PAM removed. The corresponding visualization results are presented in the first row of Fig 9 subfigure a, which demonstrates that the proposed PAM effectively highlights the most distinctive features in traditional village images. Additionally, the comparison of EPANet-KD with the de-emphasized PAM in the first and second stages further validates the efficacy of the proposed PAM, as evident in both the quantitative results and visualization outcomes.

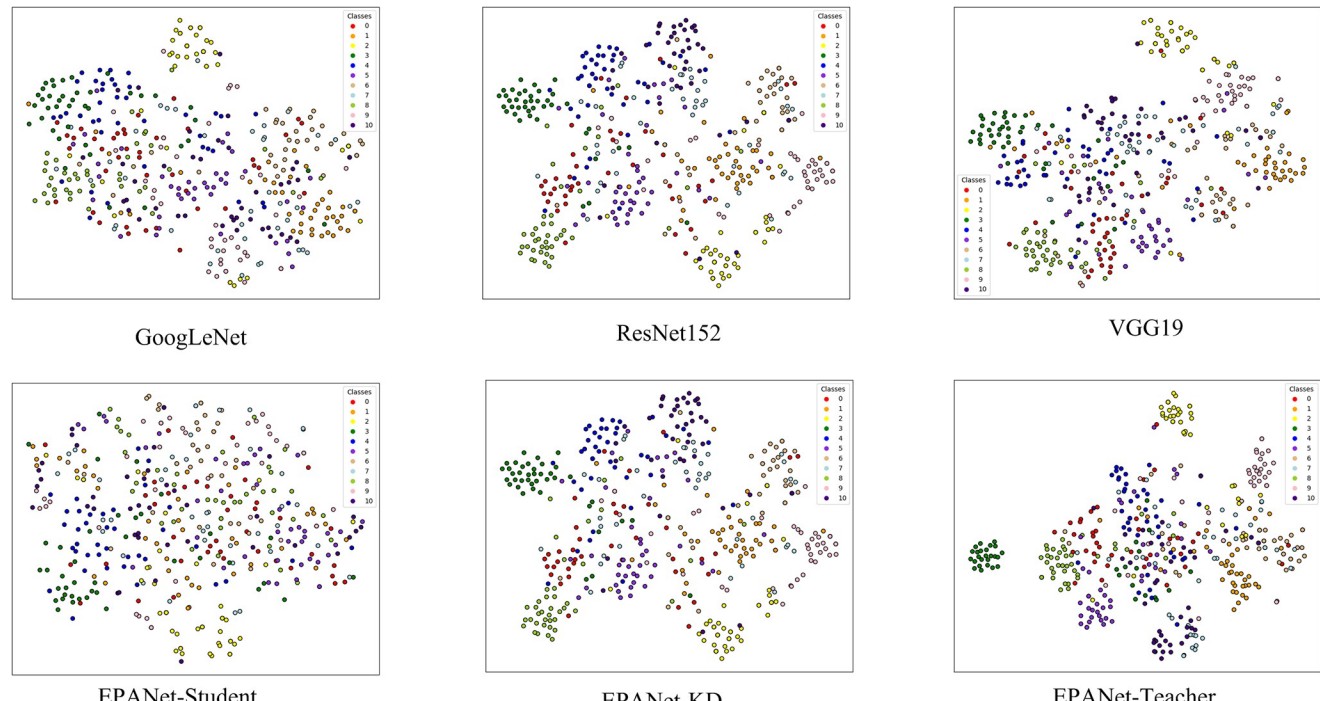

**Fig 8. t-SNE visualization comparison of fine-grained classification networks for traditional villages.**

## B. Efficiency of SAD

Traditional knowledge distillation strategies utilize complex KL and CE losses to transform knowledge from the teacher network to the student network, which is not only complex but also inefficient. In addition, the output categorization probability knowledge of the teacher network is usually transferred directly to the student network via pixel-by-pixel transfer, making it difficult for the student network to accept. To solve this problem, the SAD distillation strategy is hereby proposed, as shown in Table 3.

As shown in Table 3, W/O SAD, W/ KL and W/ CE denote removing SAD, replacing SAD with KL and replacing SAD with CE respectively on the basis of EPANet-KD. Besides, EPANet-KD improves 7.19 and 0.02, 6.13 and 0.01 relative to them in terms of Accuracy and ROC respectively, 5.45 and 0.01. In addition, the visualization results are shown in Fig 9 subfigure b, from which, it can be seen that the proposed EPANet-KD pays more attention to the most featured regions in the image, further indicating the effectiveness of the proposed SAD.

**Table 2. Comparative results of ablation experiments of PAM.**

| Method | Accuracy↑ | ROC↑ |
|---|---|---|
| EPANet-KD(W/O PAM) | 59.32 | 0.84 |
| EPANet-KD(W/O Stage1) | 60.67 | 0.84 |
| EPANet-KD(W/O Stage2) | 61.95 | 0.85 |
| EPANet-KD | 64.31 | 0.85 |

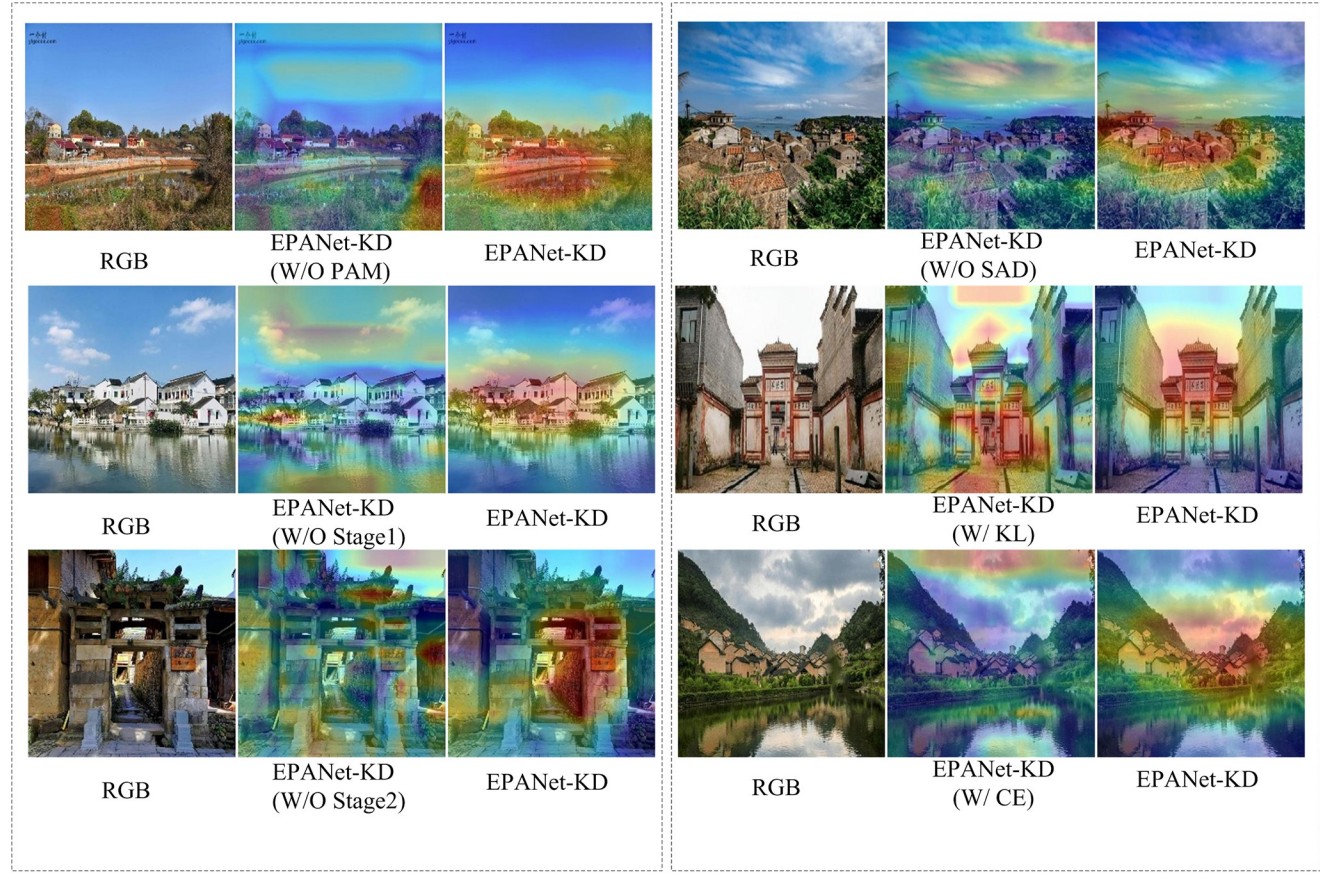

(a)Ablation visualization results of the EPANet-KD module

(b)Ablation visualization results for the EPANet-KD knowledge distillation strategy

**Fig 9. Comparison of the visualization of the ablation results of the EPANet-KD module and the knowledge distillation strategy.**

## C. Comparison of PAM with other existing attention

To further demonstrate the advantages of our proposed PAM module, we conduct quantitative and qualitative comparisons with other extant attention modules SEB [31] and SAM [32], and the results are shown in Table 4.

As shown in Table 4, W/SEB and W/SAM denote the replacement of our proposed PAM method with SEB and SAM, respectively.From the accuracy and ROC metrics, it can be seen that our proposed PAM improves the accuracy and ROC metrics in these two metrics by 4.19 and 0.04, and by 4.07 and 0.03, respectively.In addition, the visualization results are shown in Fig 10, which are different from the other methods that only focus on the spatial or channel

**Table 3. Comparative results of ablation experiments with SAD distillation strategy.**

| Method | Accuracy↑ | ROC↑ |
|---|---|---|
| EPANet-KD(W/O SAD) | 57.14 | 0.83 |
| EPANet-KD(W/ KL) | 58.18 | 0.84 |
| EPANet-KD(W/ CE) | 58.86 | 0.84 |
| EPANet-KD | 64.31 | 0.85 |

Table 4. Comparative results of PAM with other existing attention.

| Method | Accuracy↑ | ROC↑ |
|---|---|---|
| EPANet-KD(W/ SEB) | 60.12 | 0.81 |
| EPANet-KD(W/ SAM) | 60.24 | 0.82 |
| EPANet-KD | 64.31 | 0.85 |

dimensions. modeling, thanks to the fact that our PAM first performs refined attention modeling in the spatial dimension, followed by incrementally augmented modeling in the channel dimension, which results in a more complete object of attention, further illustrating the superiority of the proposed PAM.

## D. Failure cases

Although the EPANet-KD achieves comparable performance of EPANet-Teacher in Accuracy and ROC metrics, respectively, it is still subject to some limitations. As shown in Fig 11,

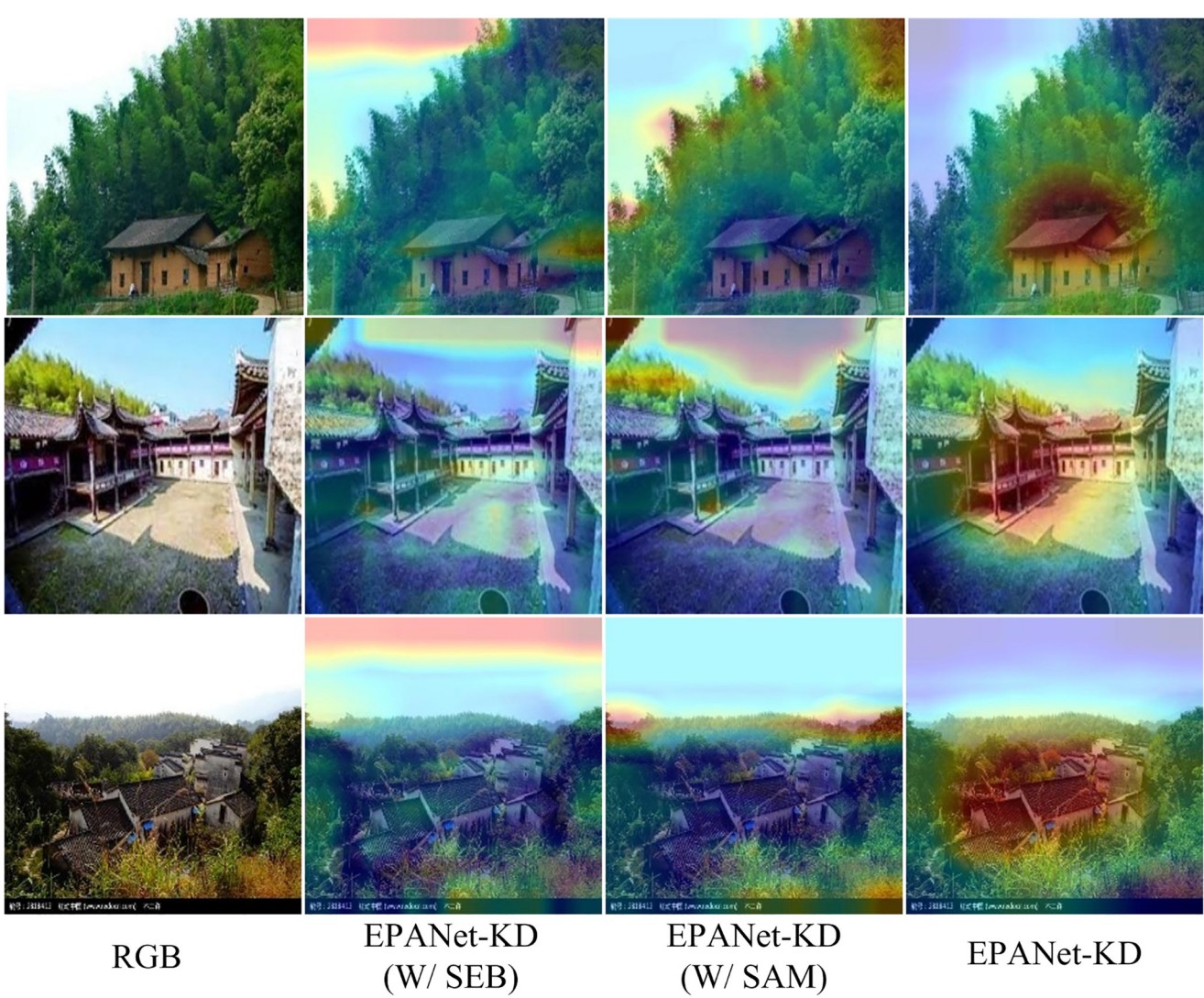

RGB          EPANet-KD          EPANet-KD          EPANet-KD
             (W/ SEB)           (W/ SAM)

Fig 10. Comparison of the visualization of PAM with other existing attention.

GT denotes the region that the image should really focus on. However, in some images, EPA-Net-KD focuses on the sky region, causing great errors in the fine-grained classification of traditional villages. The reason for this analysis is mainly a result of EPANet-Teacher transferring the learned knowledge of invalid traditional village fine-grained classification, such as the sky, to the student network. Therefore, in the future, a new more discriminative fine-grained categorization network should be proposed.

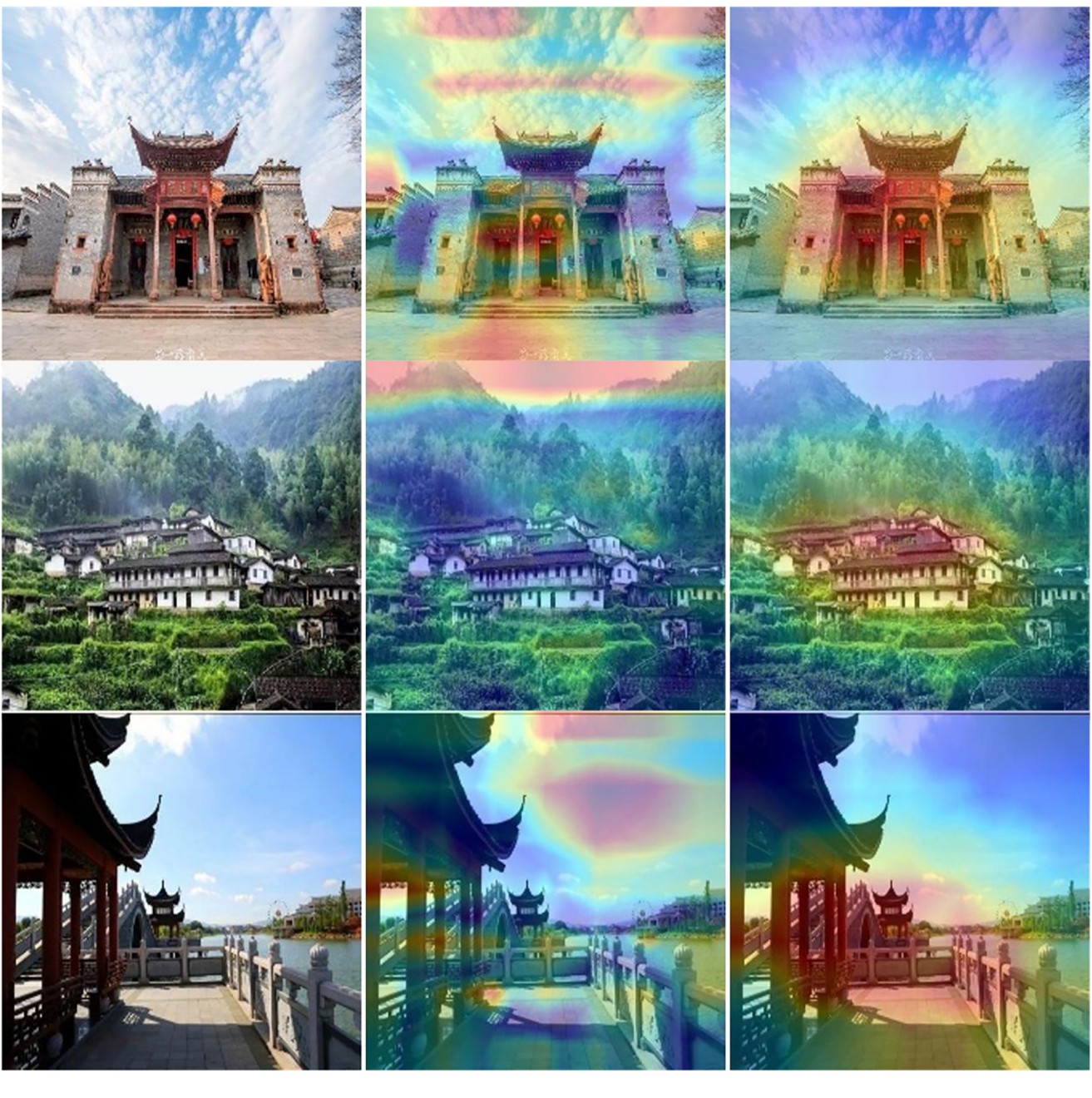

RGB             EPANet-KD             GT

**Fig 11. Visualization of failure of the EPANet-KD approach.**

## Conclusions

First, this paper provides the first large-scale fine-grained categorization dataset PVCD for the traditional village research community. Second, in order to perform more accurate and efficient fine-grained categorization on PVCD, a simple yet effective EPANet-KD is proposed, in which, a new PAM module progressively focusing on the most distinctive fine-grained categorization information in the channel and spatial dimensions is proposed, and a new SAD is proposed to achieve simple and efficient transfer of teacher network fine-grained categorization knowledge to the teacher network by softening the alignment of categorization probability distributions. In turn, it enables the proposed EPANet-KD to work well on resource-constrained devices. Finally, a comparison is made between the EPANet-KD and state-of-the-art methods, and the results show that the proposed approach achieves advanced performance. In the future, the problem of fine-grained classification of traditional villages under multimodal conditions will be further explored to promote the conservation and development of traditional villages.

## Acknowledgments

A special thank you to Jiangxi rural cultural Development Research Center for conducting and expanding on our experimental studies. All authors contributed equally to the data analysis, conducting experimentation, and overall manuscript conception.

## Author Contributions

**Conceptualization:** Cheng Zhang.

**Data curation:** Cheng Zhang.

**Formal analysis:** Cheng Zhang.

**Funding acquisition:** Cheng Zhang.

**Investigation:** Cheng Zhang, Huimin Gong.

**Methodology:** Cheng Zhang.

**Project administration:** Cheng Zhang.

**Resources:** Cheng Zhang.

**Software:** Cheng Zhang, Jinlin Teng.

**Supervision:** Cheng Zhang, Chunqing Liu.

**Validation:** Cheng Zhang, Chunqing Liu.

**Visualization:** Cheng Zhang.

**Writing – original draft:** Cheng Zhang.

**Writing – review & editing:** Cheng Zhang.

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
