## [Decision Letter · Decision Letter 0]

21 Nov 2023

PONE-D-23-36399EFPCNet:Efficient Fine-grained Provincial village

Classification Network by Attention and Knowledge

Distillation MethodsPLOS ONE

Dear Dr. Chunqing,

Thank you for submitting your manuscript to PLOS ONE. After careful consideration, we feel that it has merit but does not fully meet PLOS ONE’s publication criteria as it currently stands. Therefore, we invite you to submit a revised version of the manuscript that addresses the points raised during the review process.

We look forward to receiving your revised manuscript.

Kind regards,

Sathishkumar Veerappampalayam Easwaramoorthy

Academic Editor

PLOS ONE

Journal Requirements:

2. In your Methods section, please include additional information about your dataset and ensure that you have included a statement specifying whether the collection and analysis method complied with the terms and conditions for the source of the data.

5. Thank you for stating the following financial disclosure: "National Natural Science Foundation of China".

6. Thank you for stating the following in your Competing Interests section: "NO authors have competing interests".

7. We note that you have stated that you will provide repository information for your data at acceptance. Should your manuscript be accepted for publication, we will hold it until you provide the relevant accession numbers or DOIs necessary to access your data. If you wish to make changes to your Data Availability statement, please describe these changes in your cover letter and we will update your Data Availability statement to reflect the information you provide.

8. PLOS requires an ORCID iD for the corresponding author in Editorial Manager on papers submitted after December 6th, 2016. Please ensure that you have an ORCID iD and that it is validated in Editorial Manager. To do this, go to ‘Update my Information’ (in the upper left-hand corner of the main menu), and click on the Fetch/Validate link next to the ORCID field. This will take you to the ORCID site and allow you to create a new iD or authenticate a pre-existing iD in Editorial Manager. Please see the following video for instructions on linking an ORCID iD to your Editorial Manager account: https://www.youtube.com/watch?v=_xcclfuvtxQ

Reviewers' comments:

Reviewer's Responses to Questions

**Comments to the Author**

1. Is the manuscript technically sound, and do the data support the conclusions?

Reviewer #1: Yes

Reviewer #2: Partly

2. Has the statistical analysis been performed appropriately and rigorously? 

Reviewer #1: Yes

Reviewer #2: Yes

3. Have the authors made all data underlying the findings in their manuscript fully available?

Reviewer #1: Yes

Reviewer #2: No

4. Is the manuscript presented in an intelligible fashion and written in standard English?

Reviewer #1: Yes

Reviewer #2: Yes

5. Review Comments to the Author

Reviewer #1: In this manuscript, the authors constructed a new dataset of the first Provincial village and proposed a PAM module to focus progressively on salient objects in the channel and spatial dimensions, and a SAL distillation strategy to reduce the number of parameters and computational effort. In addition, the authors compared EFPCNet-KD network with the state-of-the-art 7 methods to obtain a more advanced performance. There are some issues that need to be addressed:

1.In Methods, does the PVCNet-KD network refer to the EFPCNet-KD? The authors should clarify the differences between these two methods.

2.This paper is not well written. There are many mistakes in spelling, punctuation, and incomplete sentences in the manuscript.

3.The figures provided in the manuscript are not clear.

4.The size of font in the tables is inconsistent. The authors should revise the format of the tables.

Reviewer #2: This article mentions the challenges of traditional village classification and identification, and an efficient attention-based fine-grained traditional village classification network (EFPCNet) is designed. This paper proposes a new knowledge distillation method with balanced probability distribution,Specifically, the probability distribution balancing loss MAE is introduced, which balances the probability distribution of the classification output of the softened EFPCNet-teacher network with the classification output of the EFPCNet-student network, as a way to transfer teacher knowledge to the student network. The results show that the method proposed in this paper achieves the best performance on the test data set. However, I think the method innovation in this article is limited, and there are some flaws and deficiencies in the method description and experiments. The specific problems are as follows.

Some suggestions are as follows:

1. The introduction discusses the importance of village research from a humanistic perspective. It does not involve relevant research on traditional villages using machine learning or deep learning methods. It does not technically emphasize the research questions and research difficulties of this article.

2. The proposed method part lacks innovation, and the description of the method part is too brief. It is more of a simple combination of existing technologies. The author did not emphasize the technological innovation and improvement part of this article.

3. Please add how to calculate the MAE in Formula 6 and the SAL and CEL losses in Formula 7.

4. The author should give some sample pictures of the data set constructed in this article.

5. The quality of all pictures in this article seriously affects the understanding of the paper work, and the author should provide higher quality illustrations.

6. The paper lacks specific explanations of some symbols in the formula, such as XxSigmoid in Formula 1 and xSigmoid in Formula 3.

7. There are some writing errors in the paper, please check carefully, such as: igh, ablation.

8. The experimental part is imperfect and lacks visual analysis of village classification.

6. PLOS authors have the option to publish the peer review history of their article (what does this mean?). If published, this will include your full peer review and any attached files.

Reviewer #1: No

Reviewer #2: No

---

## [Author Response · Author response to Decision Letter 0]

30 Dec 2023

Dear Editor,

We have studied the valuable comments from you,the assistant editor and reviewers carefully, and tried our best to revise the manuscript. The point to point responds to the reviewer's comments are listed as following:

Responds to the reviewer’ s comments:

1.Is the manuscript technically sound, and do the data support the conclusions?

The paper validates the effectiveness of the proposed method via experimentally quantitative and qualitative comparison with state-of-the-art classification methods. Meanwhile, ablation experiments are conducted on the proposed PAM and SAD to further validate the effectiveness of the proposed method. Finally, further intuitive visualization is explained with attention graphs and t-SNE.

3.Have the authors made all data underlying the findings in their manuscript fully available?

All datasets and code in the paper are publicly available from https://github.com/Jack13026212687/EPANet-KD

Reviewer #1: In this manuscript, the authors constructed a new dataset of the first Provincial village and proposed a PAM module to focus progressively on salient objects in the channel and spatial dimensions, and a SAL distillation strategy to reduce the number of parameters and computational effort. In addition, the authors compared EFPCNet-KD network with the state-of-the-art 7 methods to obtain a more advanced performance. There are some issues that need to be addressed:

1.In Methods, does the PVCNet-KD network refer to the EFPCNet-KD? The authors should clarify the differences between these two methods.

Both PVCNet-KD and EFPCNet-KD refer to EPANet-KD and have been harmonized throughout the text to all correct to EPANet-KD.

2.This paper is not well written. There are many mistakes in spelling, punctuation, and incomplete sentences in the manuscript.

We do feel sorry for the trouble. The spelling errors, punctuation and incomplete sentences have all been corrected in the manuscript.

3.The figures provided in the manuscript are not clear.

Thanks for your comment. All diagrams in the manuscript have been corrected to clear diagrams as suggested.

4.The size of font in the tables is inconsistent. The authors should revise the format of the tables.

Font sizes and formatting in all tables in the manuscript have been correspondingly corrected in all cases.

Reviewer #2: This article mentions the challenges of traditional village classification and identification, and an efficient attention-based fine-grained traditional village classification network (EFPCNet) is designed. This paper proposes a new knowledge distillation method with balanced probability distribution,Specifically, the probability distribution balancing loss MAE is introduced, which balances the probability distribution of the classification output of the softened EFPCNet-teacher network with the classification output of the EFPCNet-student network, as a way to transfer teacher knowledge to the student network. The results show that the method proposed in this paper achieves the best performance on the test data set. However, I think the method innovation in this article is limited, and there are some flaws and deficiencies in the method description and experiments. The specific problems are as follows.

Some suggestions are as follows:

1. The introduction discusses the importance of village research from a humanistic perspective. It does not involve relevant research on traditional villages using machine learning or deep learning methods. It does not technically emphasize the research questions and research difficulties of this article.

References [8-12] in the introduction section of the newly revised paper are supplemented with relevant research on traditional villages using deep learning methods. In addition, the ending sentence of each paragraph in the introduction section technically emphasizes the research problem and research difficulties of this paper.

2. The proposed method part lacks innovation, and the description of the method part is too brief. It is more of a simple combination of existing technologies. The author did not emphasize the technological innovation and improvement part of this article.

The paper constructs the first traditional village fine-grained categorization dataset PVCD. In addition, two innovations, namely PAM and SAD, are proposed. The newly revised paper provides a detailed introduction to PAM and SAD. Both the introduction and the methods section of the paper emphasize the technical innovation and improvement part of the paper.

3. Please add how to calculate the MAE in Formula 6 and the SAL and CEL losses in Formula 7.

Equation 4-9 of the newly revised paper unfolds how MAE, SAL, and CEL are calculated.

4. The author should give some sample pictures of the data set constructed in this article.

Thanks for your comment. A sample of the constructed traditional village fine-grained categorization dataset PVCD is given in the newly added Figure 2.

5. The quality of all pictures in this article seriously affects the understanding of the paper work, and the author should provide higher quality illustrations.

All figures in the full text of the new revised version have been revised to high quality.

6. The paper lacks specific explanations of some symbols in the formula, such as XxSigmoid in Formula 1 and xSigmoid in Formula 3.

Part B of the Overview in the Materials and methods section explains that it is sigmoid in PAM.

7. There are some writing errors in the paper, please check carefully, such as: igh, ablation.

Writing errors in the newly revised paper have been corrected throughout.

8. The experimental part is imperfect and lacks visual analysis of village classification.

The newly revised paper further intuitively visualizes and explains the fine-grained classification of traditional villages through attention graphs and t-SNE.

---

## [Decision Letter · Decision Letter 1]

16 Jan 2024

PONE-D-23-36399R1EPANet-KD:Efficient Progressive Attention Network for Fine-Grained Provincial Village Classification via Knowledge DistillationPLOS ONE

Dear Dr. Chunqing,

Thank you for submitting your manuscript to PLOS ONE. After careful consideration, we feel that it has merit but does not fully meet PLOS ONE’s publication criteria as it currently stands. Therefore, we invite you to submit a revised version of the manuscript that addresses the points raised during the review process.

 Please submit your revised manuscript by Mar 01 2024 11:59PM. If you will need more time than this to complete your revisions, please reply to this message or contact the journal office at plosone@plos.org. Please include the following items when submitting your revised manuscript:A rebuttal letter that responds to each point raised by the academic editor and reviewer(s). You should upload this letter as a separate file labeled 'Response to Reviewers'.A marked-up copy of your manuscript that highlights changes made to the original version. You should upload this as a separate file labeled 'Revised Manuscript with Track Changes'.An unmarked version of your revised paper without tracked changes. You should upload this as a separate file labeled 'Manuscript'.If applicable, we recommend that you deposit your laboratory protocols in protocols.io to enhance the reproducibility of your results. Protocols.io assigns your protocol its own identifier (DOI) so that it can be cited independently in the future. For instructions see: https://journals.plos.org/plosone/s/submission-guidelines#loc-laboratory-protocols. Additionally, PLOS ONE offers an option for publishing peer-reviewed Lab Protocol articles, which describe protocols hosted on protocols.io. Read more information on sharing protocols at https://plos.org/protocols?utm_medium=editorial-email&utm_source=authorletters&utm_campaign=protocols.

We look forward to receiving your revised manuscript.

Kind regards,

Sathishkumar Veerappampalayam Easwaramoorthy

Academic Editor

PLOS ONE

Journal Requirements:

Reviewers' comments:

Reviewer's Responses to Questions

**Comments to the Author**

1. If the authors have adequately addressed your comments raised in a previous round of review and you feel that this manuscript is now acceptable for publication, you may indicate that here to bypass the “Comments to the Author” section, enter your conflict of interest statement in the “Confidential to Editor” section, and submit your "Accept" recommendation.

Reviewer #1: (No Response)

Reviewer #2: (No Response)

2. Is the manuscript technically sound, and do the data support the conclusions?

Reviewer #1: Yes

Reviewer #2: (No Response)

3. Has the statistical analysis been performed appropriately and rigorously? 

Reviewer #1: Yes

Reviewer #2: (No Response)

4. Have the authors made all data underlying the findings in their manuscript fully available?

Reviewer #1: Yes

Reviewer #2: (No Response)

5. Is the manuscript presented in an intelligible fashion and written in standard English?

Reviewer #1: Yes

Reviewer #2: (No Response)

6. Review Comments to the Author

Reviewer #1: After reviewing the paper again, it seems that it has improved slightly. However, there are still some minor problems:

1.In the paragraph explaining the formula symbols in the paper, the symbols in the formula are inconsistent with the size of font in the main text.

2.The labels of the figures in the paper are inconsistent. Please check carefully.

3.Is the font in the table too small? Please modify it according to the journal's requirements.

Reviewer #2: Some suggestions are as follows:

1. The author should explain why EPANet-Teacher and EPANet-Student use different backbones, one uses Segformer-b4 and the other uses Segformer-b0.

2. This paper proposes a new attention module PAM, which should be compared with other existing attention modules to highlight the advantages and innovation of the method.

3. The latest literature review and citations are lacking. It is recommended that the author refer to the following literature.

“A collaborative gated attention network for fine-grained visual classification”

“Multi-label image classification with multi-scale global-local semantic graph network”

“Mining graph-based dynamic relationships for object detection”

7. PLOS authors have the option to publish the peer review history of their article (what does this mean?). If published, this will include your full peer review and any attached files.

Reviewer #1: No

Reviewer #2: No

---

## [Author Response · Author response to Decision Letter 1]

18 Jan 2024

We tried our best to improve the manuscript and made some changes to the manuscript. These changes will not influence the content and framework of the paper. And here we did not list the changes but marked in red in the revised paper. We appreciate for Editors/Reviewers’ warm work earnestly and hope that the correction will meet with approval.

---

## [Editor Report · Decision Letter 2]

25 Jan 2024

EPANet-KD:Efficient Progressive Attention Network for Fine-Grained Provincial Village Classification via Knowledge Distillation

PONE-D-23-36399R2

Dear Dr. Chunqing,

We’re pleased to inform you that your manuscript has been judged scientifically suitable for publication and will be formally accepted for publication once it meets all outstanding technical requirements.

Kind regards,

Sathishkumar Veerappampalayam Easwaramoorthy

Academic Editor

PLOS ONE
---

## [Editor Report · Acceptance letter]

31 Jan 2024

PONE-D-23-36399R2 

PLOS ONE

Dear Dr. Liu, 

I'm pleased to inform you that your manuscript has been deemed suitable for publication in PLOS ONE. Congratulations! Your manuscript is now being handed over to our production team.

Kind regards, 

on behalf of

Dr. Sathishkumar Veerappampalayam Easwaramoorthy 

Academic Editor

PLOS ONE